

# An ensemble method for improving the estimation of planetary boundary layer height from radiosonde data

Xi Chen[1,2], Ting Yang[1*], Zifa Wang[1], Futing Wang[1,3], Haibo Wang[1,3]

[1]State Key Laboratory of Atmospheric Boundary Layer Physics and Atmospheric Chemistry (LAPC), Institute of Atmospheric Physics, Chinese Academy of Sciences, Beijing 100029, China
[2]Jiangsu Key Laboratory of Atmospheric Environment Monitoring and Pollution Control, Jiangsu Collaborative Innovation Center of Atmospheric Environment and Equipment Technology, School of Environmental Science & Engineering, Nanjing University of Information Science & Technology, Nanjing, 210044, China
[3]University of Chinese Academy of Sciences, Beijing 100049, China

*Correspondence to*: Ting Yang (tingyang@mail.iap.ac.cn)

**Abstract.** The planetary boundary layer (PBL) height (PBLH) is an important parameter for both weather, climate and air quality models. Radiosonde is one of the commonly used instruments for PBLH determination and is generally accepted as a standard for other methods. However, mainstream approaches for the estimation of PBLH from radiosonde present some uncertainties and even show disadvantages under some circumstances and the results need to be visually verified, especially during the transition period of different PBL regimes. To avoid the limitations of individual methods and provide a benchmark estimation of PBLH, we propose an ensemble method based on high-resolution radiosonde data collected in Beijing in 2017. Seven existing methods including four gradient-based methods are combined along with statistical modification. The ensemble method is verified during afternoon, morning, and evening transition periods, respectively. The overestimation of PBLH can be effectively eliminated by setting threshold for gradient-based methods and the inconsistency between individual methods can be reduced by clustering. Based on the statistics of one-year observational analysis, the effectiveness of the ensemble method reaches up to 70.8%, an increase of 14.7% ~ 61.2% compared with the existing methods. Nevertheless, the ensemble method suffers to some extent from uncertainties caused by the removal of truly high PBLH, the profiles with a multi-layer structure, and the intermittent turbulence in the stable boundary layer (SBL). Finally, this method has been applied to characterize the diurnal and seasonal variations of different PBL regimes. Particularly, the average CBL height is found to be the highest in spring and the SBL is lowest in summer with about 200 m. The average PBLH at transition stage lies around 900 m and there is no obvious seasonal variation. The findings imply the effectiveness of the ensemble method.



# 1 Introduction

The planetary boundary layer (PBL) has shown to exert significant impact on the diurnal variation of key meteorological variables, due to its close contact with ground surface (Wallace and Hobbs, 2006). In the daytime under fair weather conditions, solar heating causes the rapid development of a convective boundary layer (CBL) (Oke, 1988). Conversely, the infrared radiative cooling after sunset results in the generation of a stable boundary layer (SBL) with a residual layer (RL) of the daytime CBL aloft (Fernando and Weil, 2010), even though this evening transition is slow (Yuval et al., 2020). As a

fundamental variable to characterize the structure of PBL, the PBL height (PBLH) reflects the vertical extent of turbulent mixing at which the exchanges with the free troposphere take place (Seidel et al., 2010). The PBLH is a key parameter for weather, climate, and air-pollution models to describe many critical tropospheric processes, such as convective transport, cloud entrainment, and pollutant diffusion (Liao et al., 2015; Liu and Liang, 2010; Lou et al., 2019; Seibert et al., 2000). Therefore, the estimation of PBLH has aroused wide interest.

Recently, many algorithms have been developed to get an explicit estimation of PBLH from various instruments, especially from remote-sensing instruments. Ground-based lidar is one commonly used instrument for continual retrieval of PBLH by tracking the gradient of aerosol backscatter. Gradient-based algorithms (Yang et al., 2017; Flamant et al., 1997; Summa et al., 2013; He et al., 2006) and wavelet transform algorithms (Davis et al., 2000; Baars et al., 2008; Brooks, 2003) are two main categories of lidar algorithms. The combination of different algorithms (Zhang et al., 2020; Sawyer and Li, 2013; Zhang et al.,

2022), the introduction of image processing (Vivone et al., 2021; Lewis et al., 2013), the extended-Kalman filtering technique (Kokkalis et al., 2020), and the consideration of stability (Su et al., 2020) are all new directions in the evolution of lidar algorithms. Radar wind-profiler is another active remote sensor to obtain PBLHs. Huang et al. (2016) combined the threshold method and the fractional method to get more reliable diurnal cycle of PBLH from radar wind profiler. Liu et al. (2019) developed an improved threshold method using normalized signal-to-noise ratio profiles. Besides, the algorithm used to

retrieve PBLH from ceilometer is similar to that from lidar, and new algorithms emerge continuously (Kambezidis et al., 2021; Eresmaa et al., 2006; Min et al., 2020). Among the above-mentioned instruments, radiosonde is a traditional and reference instrument for the verification of the reliability of new algorithms, so the accuracy of radiosonde results is crucial.

For the estimation of PBLH from radiosonde, determination of the rapid change in temperature, air-moisture, and wind-direction profiles is a well-established and the most commonly used approach (Seibert et al., 2000). Other approaches, such as

the "parcel method", which uses a hypothetical parcel of air as a thermal (Holzworth, 1964), and the bulk Richardson number method (Vogelezang and Holtslag, 1996) have been widely used in various researches as well. However, each method has certain limitations and there is no individual method that can be applied under all atmospheric conditions (Li et al., 2021). The surface-based inversion (SI) method can only be considered under stable conditions, while the parcel method works under convective atmospheric conditions. The methods that relied on humidity information usually suffer from the changing

measurability of humidity, the existence of clouds, and the measurement error of humidity instruments (Wang and Wang, 2014). Although often used, the bulk Richardson number method has limitations like the determination of the threshold (Seidel





et al., 2012), the low detection rate of SBL (Dai et al., 2014), and the too-shallow CBL with greater static stability (Lemone et al., 2013). Recent research also suggests that bias of thermodynamic or kinematic fields in the bulk Richardson number method could introduce PBLH errors of ~300 m (Lee and Pal, 2021). Moreover, different methods show inconsistent results. Seidel et al. (2010) compared the PBLH derived from seven existing methods and found several hundred meters of differences among them. Such differences can also be found in the cases presented in Kambezidis et al. (2021), and visual validation is still the most reliable.

To address the inconsistency between different elements and the existence of clouds, the methods that integrated the potential temperature, relative humidity, specific humidity, and atmospheric refractivity methods were proposed (Wang and Wang, 2014), which tended to generate a more consistent estimation of PBLH. Another objective method of collocating the virtual potential temperature with the dewpoint was designed in an attempt to decrease the errors of PBLH (Niyogi and Schmid, 2012). Nevertheless, the above-mentioned integrated methods are mostly confined to daytime observations, ignoring the determination of PBLH during the transition period. As the routine radiosonde generally operates twice a day at 0800 and 2000 Beijing Time (BJT = UTC+8), the sounding data in China correspond to the local morning or evening when the PBL is changing rapidly. These data in the transition have more complex PBL structures than the convective boundary layer (CBL) in the daytime. To obtain high quality PBLH results, a customized data processing is required (Kotthaus et al., 2023). Previous studies have mentioned that further understanding of the transition period will benefit the model development for both meteorology and air pollution applications (Jensen et al., 2015; Taylor et al., 2014). Thus, it is imperative to combine multiple methods to improve the accuracy of PBLH estimation without visual validation, especially for transition periods.

Herein, to obtain a consistent and accurate estimation of the PBLH from radiosonde for the validation of other instruments, we propose an ensemble method that combines seven individual standards to avoid the limitations of individual methods. Then, we apply it to the high-resolution radiosonde data from a site of the China Radiosonde Network (CRN), which is usually operated during the transition period. Section 2 provides a brief description of the data and existing methods, our methodology, and the classification of PBL regime. The effectiveness and uncertainty of the ensemble method are presented in Section 3, followed by a summary and discussion in Section 4.

## 2 Methods and Data

### 2.1 Radiosonde Data

The GTS1 digital electronic radiosonde data are obtained from the routine observations of the L-band China Radiosonde Network (CRN) conducted by the China Meteorological Administration (CMA). Generally, this data set can provide high temporal resolution (1 s) profiles of five main meteorological elements, including temperature, relative humidity, pressure, wind speed, and direction, with an average vertical resolution of 5-8 m below 3000 m, and the data is collected twice daily at 0800 and 2000 BJT. In the summertime, intensive observations are carried out occasionally at 1400 and 0200 BJT at selected



stations (Guo et al., 2016b). Previous studies (Bian et al., 2011; Moradi et al., 2013) have proved the data accuracy of CRN is comparable with GPS radiosonde measurements produced by Vaisala.

One-year data collected in 2017 at the Beijing weather station (39.48°N, 116.28°E), which is an operational radiosonde station located in the North China Plain (NCP) with an altitude of 34 m, are used here to obtain the PBLH. The regular observations correspond to the local time at 0800 and 2000 BJT, during which the PBL is usually in the transition rather than the active convection period. A total of 817 profiles are analyzed, including 362 profiles at 0800 BJT and 361 profiles at 2000 BJT for the transition, and 94 profiles at 1400 BJT for the mature period.

**2.2 Existing Methods of PBLH Estimation**

Various methods based on radiosonde data have been developed to estimate PBLH (Seibert et al., 2000), seven widely accepted methods including both subjective and objective methods are briefly summarized below. The potential temperature ($\theta$) method defined the level of the maximum vertical gradient of $\theta$ as PBLH. Three additional gradient-based methods which assumed the PBL as a moister, denser, or more refractive layer (Seidel et al., 2010) estimate the PBLH as the level of the minimum
vertical gradient of specific humidity ($q$), relative humidity ($RH$), or refractivity ($N$), respectively. The refractivity was given by

$$N = 77.6(\frac{p}{T}) + 373000(\frac{e_v}{T^2})$$
,                                                                                        (1)

where $p$ is atmospheric pressure, $T$ is the atmospheric temperature, and $e_v$ is the water vapor pressure. The base of an elevated temperature inversion (EI) is often defined as the PBLH under convective conditions, while the top of a surface-based inversion
(SI) only can be considered as the PBLH for SBL. These two methods were not executed simultaneously, and all the results marked on temperature profiles herein are derived from one of them. The bulk Richardson number ($Ri$) method has been commonly used for radiosonde data (Seidel et al., 2012), and $Ri$ is expressed as

$$Ri(z) = \frac{(g / \theta_{vs})(\theta_{vz} - \theta_{vs})(z - z_s)}{(u_z - u_s)^2 + (v_z - v_s)^2 + (bu_*^2)}$$
,                                                                            (2)

where $z$ is the height, $s$ represents the surface, $g$ is the gravity acceleration, $\theta_v$ is the virtual potential temperature, $u$ and $v$ are
the streamwise and cross-stream wind speeds, and $u_*$ is the surface-friction velocity that generally is ignored for calculation due to the much smaller magnitude compared with other terms. We defined the PBLH as the lowest level at which $Ri$ crosses the critical value of 0.25. The criteria are referred to the previous applications in China (Guo et al., 2016b).

**2.3 An Ensemble Method**

As the PBLH values derived from existing methods show obvious disagreement and misjudgment, we proposed an ensemble
method to automatically obtain a consistent estimation of PBLH in the transition. Seven methods introduced in the previous section are included in the ensemble method to make up for the shortcomings of a single approach and to make it more suitable


for different types of boundary layers. For all methods, to avoid taking tropospheric features as the PBL top, we restricted the radiosonde data to 3000 m according to the long-term climatology of PBLH in China (Guo et al., 2019; 2021). To avoid noisy readings near the surface (Liu and Liang, 2010), we only consider the stable boundary layer higher than 100 m a.g.l. Besides

original data, three-point smoothing was also introduced to eliminate the fluctuations in high-resolution data and more details are illustrated in the supplement. Based on these constraints, the procedures of our ensemble method are specifically described as follows:

1. Apply the seven individual methods to both the raw data and the smoothed data, respectively, which are illustrated in Fig.1. Notice that the EI method will only be implemented when the result of the SI method is null.

2. Modify the results of gradient methods ($\theta$, $RH$, $q$, and $N$). Get the 75% quantiles of initial results of gradient methods from the one-year observations at 0800/2000 BJT, respectively. Compare the result of each gradient method with the corresponding 75% quantile. If the initial result is greater than the corresponding 75% quantile, then go through altitudes of the 10 smallest (or largest for $\theta$) gradients and replace the initial result with the first altitude less than 75% quantile. If all altitudes do not meet the criteria, then the PBLH for the specific observation is null.

3. Group the PBLH data set after modification by 50 m. Determine the average of the group with the largest data volume as the ensemble PBLH ($h_{ens}$). If the result of the SI method is included in this group, then take it as $h_{ens}$.

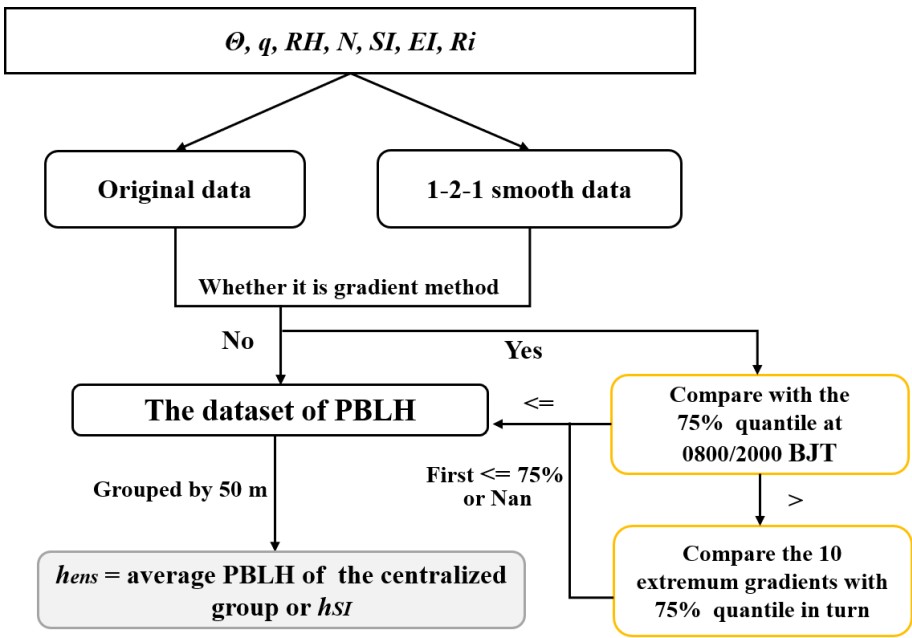

**Figure 1: The flow chart of the ensemble method.**

The result of each step at the specific observation time (0800 BJT on 15 January 2017) is illustrated as follows for further

explanation. The original results of seven individual methods at step 1 are presented in Table 1. As the potential temperature ($\theta$), specific humidity ($q$), relative humidity ($RH$), and refractivity ($N$) methods can be thought of as gradient methods, statistical modification was carried out. As this case was observed at 0800 BJT, the results were compared with the 75% quantiles of





each method at 0800 BJT, which were calculated based on the whole year of observation. Here, the 75% quantiles of each method are 1944 m, 1473 m, 1824 m, and 1346 m for the original data, and 1959 m, 1599 m, 2110 m, and 1516 m for the

smoothed data, respectively. The PBLH calculated from the smoothed data by $\theta$ method is greater than the corresponding 75% quantile, so we went through the altitudes of the 10 largest gradients and replaced the result by the first altitude smaller than the 75% quantile. The altitude (1874 m) of the third gradient met this criterion, and 1994 m was replaced by it. All other PBLHs greater than the corresponding 75% quantile were also replaced. None of the 10 altitudes from the $N$ method met this criterion, thus the result was set to be null.

**Table 2: The PBLHs (m) determined by seven methods in step 1 and step 2 at 0800 BJT on 15 January 2017.**

|  |  | $\theta$ | $q$ | $RH$ | $N$ | SI | EI | $Ri$ |
|---|---|---|---|---|---|---|---|---|
| Step 1 | Original data | 1874 | 1936 | 1362 | 1998 | null | 1362 | 61 |
|  | Smooth data | 1994 | 1936 | 1936 | 1994 | null | 1358 | 61 |
| Step 2 | Original data | 1874 | 1375 | 1362 | null | null | 1362 | 61 |
|  | Smooth data | 1874 | 1358 | 1936 | 1375 | null | 1358 | 61 |

Finally, the dataset in step 2 was grouped by 50 m, and the groups are shown in Table 2. As Group 2 has the maximum data volume, the average of Group 2, 1365 m, was taken as the ensemble PBLH in this case. If the result from the SI method is

included in the most centralized group under other circumstances, the result from the SI method will be taken as the ensemble PBLH.

**Table 2: The PBLHs (m) groups in step 3 at 0800 BJT on 15 January 2017.**

| Group 1 | 61, 61 |
|---|---|
| Group 2 | 1358, 1358, 1362, 1362, 1375, 1375 |
| Group 3 | 1874, 1874 |
| Group 4 | 1936 |

### 2.4 Classification of PBL Regimes

CBL, stable boundary layer (SBL), and residual layer (RL) are three major regime of PBL in view of its thermodynamic condition (Stull, 1988; Zhang et al., 2018). The CBL usually occurs in the daytime, and the SBL during nighttime because of the dominance of outgoing longwave radiation emitting from the ground surface. During the transition stage between CBL and SBL, the PBL structure can be complex, including the neutral RL started from the ground surface with no evident CBL or SBL (Fig. 2b), weak convective layer, and weak stable layer with RL located at the top. Herein, we define the above-mentioned

PBL regimes as transition stage (TS). The criteria of classifying the PBL regimes by examining the near-surface thermal gradient were referred to Liu and Liang (2010) and some parameters were modified according to the actual application. Figure





2 illustrates three real atmosphere cases for each PBL regime. The potential temperature difference ($\Delta\theta$) between level k (the first level right above 100 m ground level) and level 2 (the second received data) was applied as the diagnostic quantity. Through visual validation, the limit of $\Delta\theta$ is - 0.2 k for CBL, 1 k for SBL, and all other cases are classified as TS.

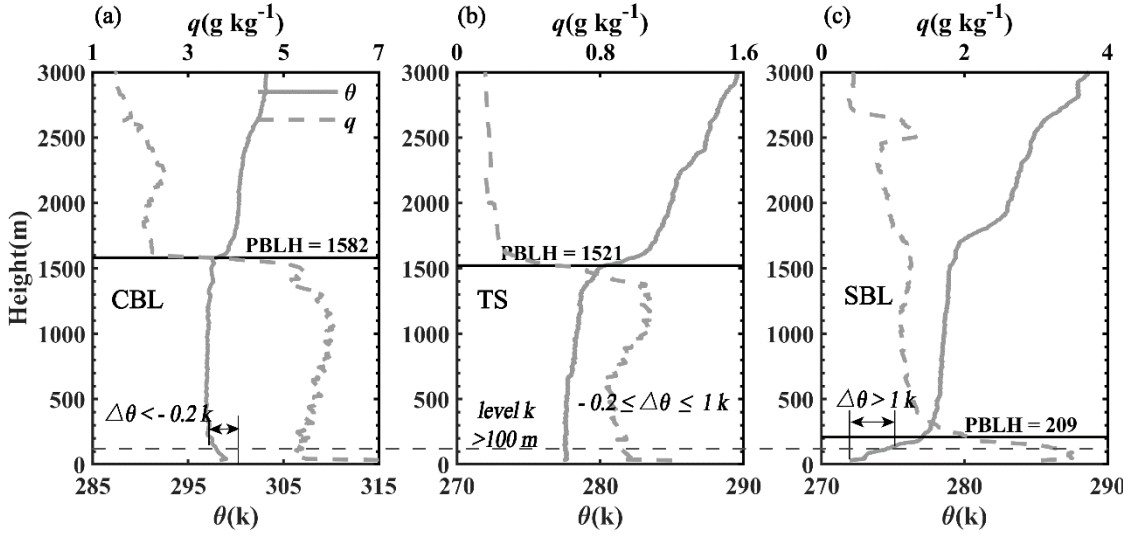

**Figure 2: The PBL structure from cases (a) at 1400 BJT on 3 June 2017 for CBL, (b) at 0800 BJT on 12 March 2017 for TS, and (c) at 2000 BJT on 8 January 2017 for SBL, respectively. The procedure of PBL classification is illustrated by the annotation ($\Delta\theta = \theta_k - \theta_2$)**

## 3 Results

### 3.1 Valid Cases under Different Conditions

The effectiveness of the ensemble method is first visually illustrated by several cases under different conditions. To assist the verification, the Range-Squared-Corrected-Signal (RSCS) at 1064 nm from a ground-based lidar located in the Institute of Atmospheric Physics (IAP), Chinese Academy of Sciences (CAS) (39.982°N, 116.385°E) is shown along with the radiosonde profiles. More information about the ground-based lidar can be found in Wang et al. (2020). The PBLHs determined by the widely used gradient method (GM), which takes the position of the minimum gradient of RSCS as the PBLH (Flamant et al., 1997), are also marked on the lidar signal profiles.

### 3.1.1 A case in the afternoon

Since the PBLH definition for the CBL is relatively clear and previous integrated methods are mainly concerned with this condition, our ensemble method was first evaluated when a CBL occurs to prove the reliability preliminarily. The observations in summer were intensified for improving the accuracy of severe weather forecasting, resulting in an additional sounding at 1400 BJT, which usually corresponds to vigorous CBL. Thus, a case at 1400 BJT on 14 June (Fig. 3) was chosen. As illustrated

in Fig. 3a, all methods initially determined the PBLH at 2859 m and that triggered the statistical modification of overestimation from gradient-based methods. There is a distinct super adiabatic layer in the profile of potential temperature, which implies the existence of CBL. The development of CBL is a continuous process promoted by the warming of solar radiation after

sunrise. From Fig. 3b, we can see that the boundary layer gradually developed from 300 m at 0800 BJT to 1650 m at 1530 BJT. Therefore, a sudden increase of PBLH to approximately 3000 m is obviously unreasonable. After the modification of gradient-based methods, the ensemble method finally determined the PBLH at 1000 m, which is 200 m less than the PBLH retrieved by GM. The boundary layer developments between these two sites are not completely synchronous due to the distance of dozens of kilometres. Hence, we affirmed the result of the ensemble method is valid and the elimination of abnormally high

PBLH in the step 2 works.

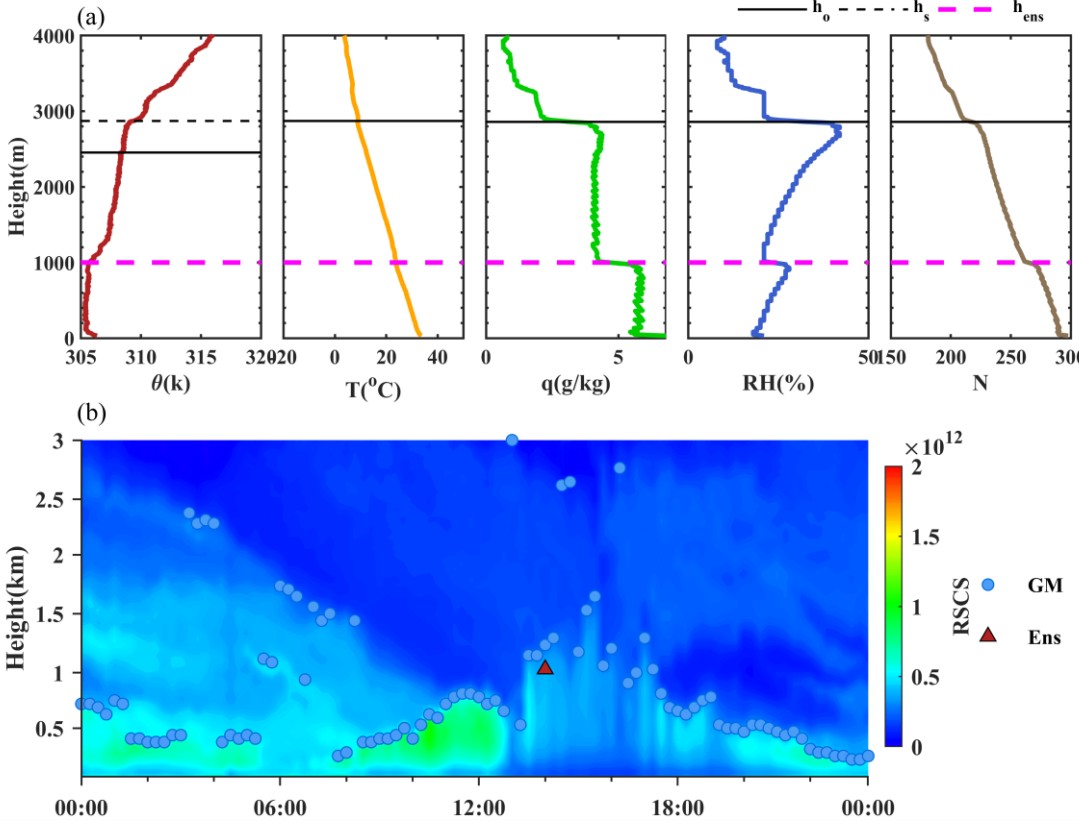

**Figure 3:** (a)The profiles of potential temperature ($\theta$), temperature ($T$), specific humidity ($q$), relative humidity ($RH$), and refractivity ($N$) at 1400 BJT on 14 June 2017. $h_o$ is the PBLH determined by original data, $h_s$ is the PBLH determined by data with three-point smoothing, and $h_{ens}$ is the PBLH determined by the ensemble method. (b) Evolution of the lidar RSCS signal at 1064 nm on 14 June
2017. The PBLHs retrieved from gradient method (GM) are marked by bule dots and the $h_{ens}$ in (a) is marked by a red triangle.

**3.1.2 A case study during the morning transition period**

During the morning TS, the top of RL continues to collapse with the infrared radiative cooling overnight. On the other hand, the CBL started to grow after sunrise. A case on 29 January with a distinct RL collapse process is shown in Fig. 4b. We can



clearly identify from the lidar RSCS signal profile that the PBLH decreased from approximately 1500 m to 900 m in the early

morning and it is 1230 m at 0800 BJT. According to the radiosonde profiles, there is no super adiabatic or inversion layer near

the surface and this case can be classified as being neutral RL. Different from the consistent overestimation in Fig. 3, the

PBLHs of each method vary greatly from 1156 m to 2768 m, among which the $q$ method is the highest and the $RH$ method is

the lowest. This kind of divergence usually requires manual verification to determine the final PBLH. However, with the

clustering in the ensemble method, the outlier results from $\theta$, $q$, $RH$ methods were automatically eliminated and the PBLH was

determined at 1175 m. The PBLH from the ensemble method shows a good consistency with the continuous boundary layer

collapse from the lidar signal profile.

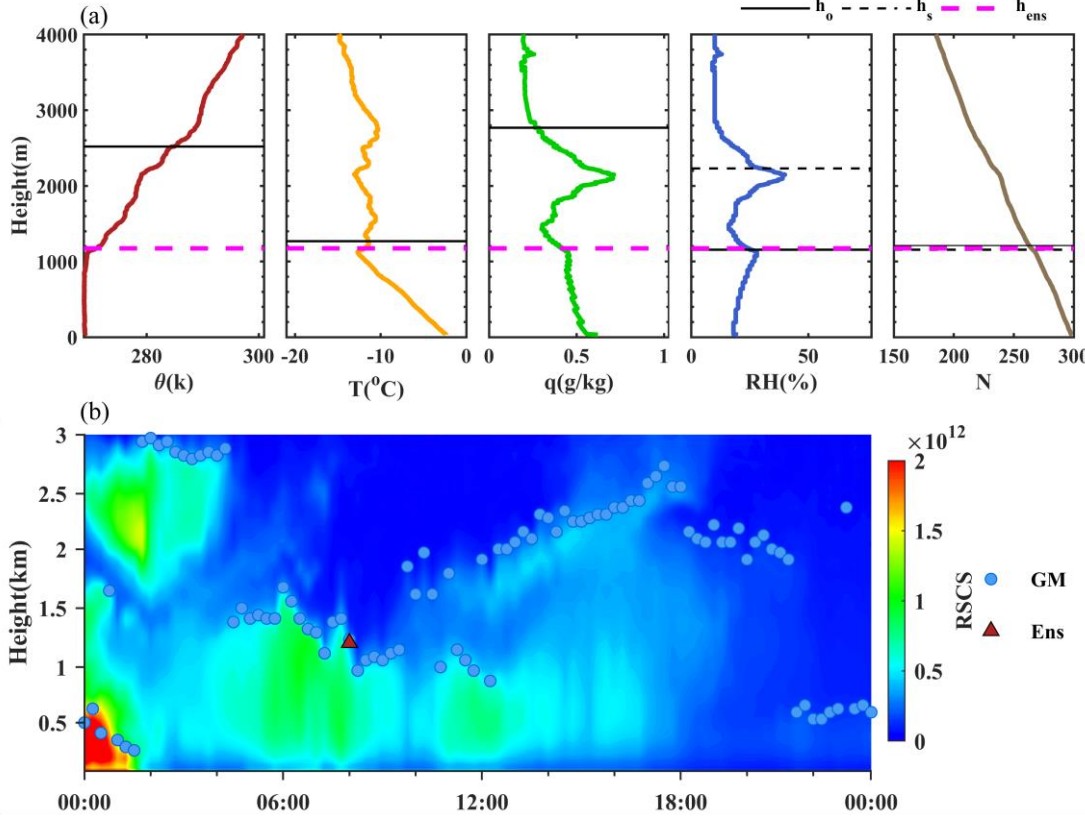

**Figure 4: Same as Fig. 3 but (a) at 0800 BJT on 29 January 2017, (b) on 29 January 2017.**

### 3.1.3 A case study during the evening transition period

Compared with morning transition, the evening transition period can be longer. Fig. 5 presents a case during evening transition

period with multi-layer structure to illustrate the applicability of the ensemble method in complex boundary layer structure.

According to the satellite cloud product and the humidity profiles of radiosonde, there were multi-layer clouds on 5 January.

The complex temperature and humidity profile caused the misestimation of PBLH at approximately 1600 m by $\theta$ and $RH$

methods. The air quality was poor, and the aerosols gathered below PBLH on that day. In this case, the lidar RSCS signal



profile shows the position of PBLH well. Constrained by the clouds, the warming effect of solar radiation after sunrise was weaker than the previous two cases and the maximum PBLH developed to 1000 m at 1200 BJT. On the other hand, the presence of clouds also slowed down the radiative cooling at night, the PBLH decreased slowly from 900 m (1730 BJT) to 600 m (2000 BJT) after sunset. Obviously, the PBLH over 1000 m from $\theta$ and $RH$ methods during evening transition is not consistent with the evolution of PBL. The ensemble method excluded the outliers of the PBLH and determined the PBLH at 536 m, which is

similar to the inversion results from lidar.

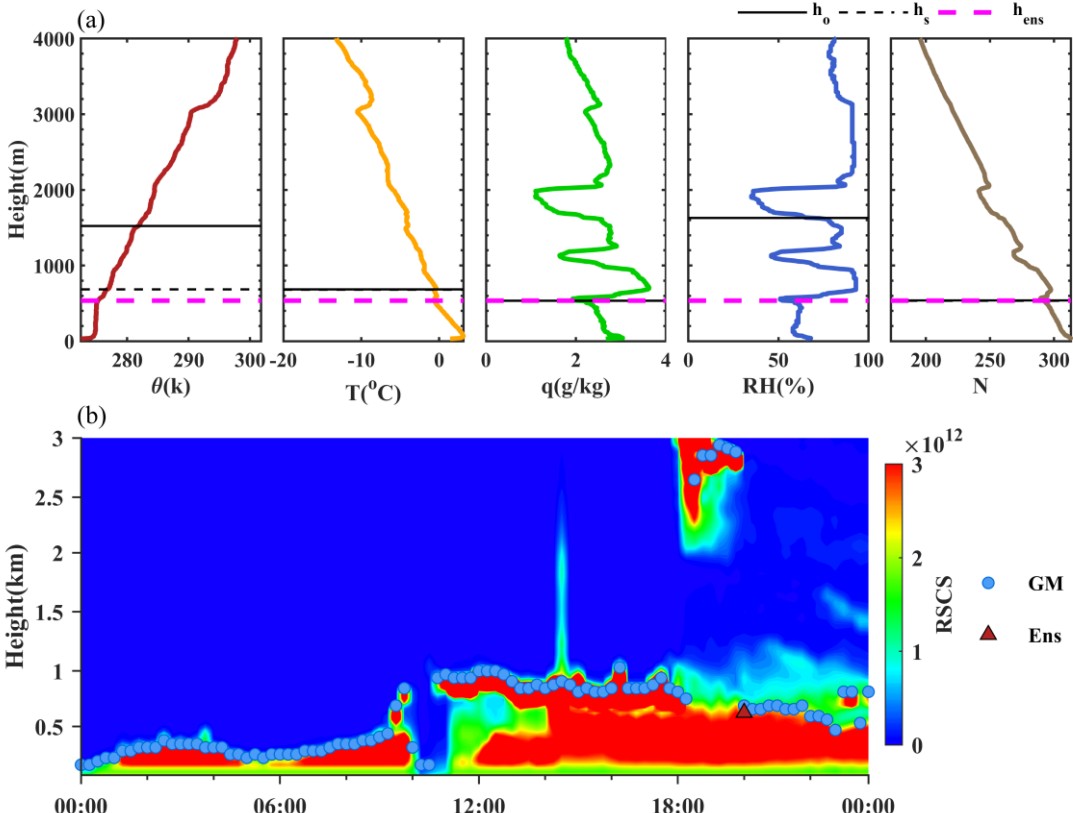

**Figure 5: Same as Fig. 3 but (a) at 2000 BJT on 5 January 2017, (b) on 5 January 2017.**

**3.2 Effectiveness and Uncertainty of the Ensemble Method**

Several cases discussed above have already shown the effectiveness of the ensemble method and in this section, we will further

prove the effectiveness and reliability of the ensemble method by statistical approaches. First, we compared the statistical characteristics with the existing methods. Table 3 shows the average PBLHs determined by seven existing methods and the ensemble method at two routine observation times. Note that all PBLHs estimated by individual methods are derived from original data in statistical analysis. The average values of PBLH are between 150 and 1600 m. The PBLHs at 0800 BJT are consistently lower than that at 2000 BJT except for the SI method, as longer surface-radiation cooling contributes to the further

development of SBL. The average PBLH from the ensemble method shows to be 676 and 917m at two routine launch times,





respectively. Comparing the other methods, we found the $\theta$ method yields consistently higher PBLH and the results are more discrete (Fig. S3). The PBLHs based on the *RH* gradient are also consistently high and that may be explained by the sensitivity of *RH* profiles to cloud-top heights (Seidel et al., 2010). The EI method shows good consistency with the integrated method at 0800 BJT with an average of 750 m PBLH. However, the *N* gradient method gave closer average PBLHs to the integrated

method at both times, with the highest correlation coefficient of 0.56. Moreover, the PBLHs derived from the *Ri* method were systematically underestimated with an average of about 300 m, which is comparable with the average nighttime PBLH in China (Guo et al., 2016a). The average PBLHs and correlation analysis with existing methods preliminarily show the rationality of the ensemble method. Meanwhile, the drawbacks of different methods are also on display.

**Table 3: The average PBLHs (m) determined by eight methods at two routine observation times and the correlation coefficient (*R*)**

**of individual methods with the integrated method.**

|  | θ | EI | SI | q | RH | N | Ri | Int |
|---|---|---|---|---|---|---|---|---|
| 08 BJT | 1153 | 750 | 199 | 975 | 1096 | 887 | 272 | 676 |
| 20 BJT | 1564 | 1384 | 163 | 1163 | 1362 | 1090 | 338 | 917 |
| *R* | 0.27[*] | 0.32[*] | -0.10 | 0.54[*] | 0.38[*] | 0.56[*] | 0.37[*] | - |

[*]Data passed the significance test

To further illustrate the effectiveness of the ensemble method, the PBLH results of each case were verified manually with the aid of lidar observation. If the error between the result of the corresponding method and the truth PBLH value is within 50 m, then the result will be considered valid. We measured the effectiveness of each method over a year of observations and the

results of existing methods are still derived from original data. The *Ri* method shows the lowest effectiveness of 9.6% and this can also be inferred from the underestimated PBLH in Table 3. As SI and EI method only be executed under specific conditions, the effectiveness of them is also low. Among four gradient methods, the $\theta$ method has an effectiveness of 33.9% and the others have higher effectiveness. They are 51.2%, 48.0%, and 56.1% for the *q*, *RH*, and *N* methods, respectively. Compared with these methods, the effectiveness of the ensemble method (70.8%) has been significantly improved.

Despite the good performance of the ensemble method, there is still some uncertainties in the calculation process. The upper quartile was chosen as the threshold in step 2 as it is widely used in statistics to get a reasonable data range. In step 2, the proportion of that all 10 altitudes are greater than 75% quantile is 2.8%, 3.9%, 4.4%, and 2.3% for the $\theta$, *q*, *RH*, and *N* methods, respectively. We also suggest getting the climatology value from the previous research on the climatology of PBLH for practical application. Apparently, the different thresholds could derive different PBLHs. To illustrate the uncertainty of PBLHs

discussed in this paper, we compared the average PBLHs of different PBL regimes using 70%, 75%, and 80% quantiles as the threshold in step 2. The higher the threshold is, the higher the average PBLH is. The average PBLHs of all observations are 799 m, 856 m, and 925 m for corresponding thresholds and there were about 12.5% (103) cases in which the PBLH changed by more than 100 m. In addition, the PBL in the TS was most affected and the SBL was least affected by the threshold. The mean PBLH of TS both increased by approximately 75 m when the threshold increased from 70% to 75% quantile and from





75% to 80% quantile. However, the difference of mean PBLH of SBL between 80% and 75% quantile (53 m) is twice that between 75% and 70% quantile (27 m), which means that a higher threshold could cause greater uncertainty.

**3.3 Invalid Cases under Different Conditions**

As the goal of the ensemble method is to improve the accuracy of automatic PBLH estimation as much as possible, it does not mean that there are no failures. In some conditions, the ensemble method fails. We will discuss the typical invalid cases in this
section.

Since the 75% quantile is applied for the modification in the ensemble method, the underestimation of PBLH caused by the removal of truly high PBL is one of the important reasons for the failure of the ensemble method. In all cases, this can account for 1.3%. In Fig. 6, all existing methods determined the PBLH at approximately 2200 m and this height exceeded the 75% quantile for $q$, $RH$ and $N$ methods. The statistical modification was initiated, which resulted in the underestimation of the
PBLH as 250 m. However, there was no evident surface-based inversion layer in the radiosonde profiles and the PBL can be classified as neutral RL. With the collapse of the PBL after sunset, the top of RL gradually descended from a maximum of about 2500 m to 2200 m at 2000 BJT and the ensemble method failed. Li et al. (2021) pointed out that the structure of the boundary layer will affect the reliability of the PBLH results. The ensemble method is also easy to fail when the profiles have a multi-layer structure and the divergence between different methods is substantial. A case (at 2000 BJT on 14 January) with
four moister layers in the profiles of humidity is shown in Fig. 7. The PBLH determined by different methods were dispersed at four layers and the scattered results led to the failure of the ensemble method to obtain the true PBLH by dominant grouping. The ensemble method mistook the first moister layer (851 m) as PBLH, while the PBLH should be 1375 m according to the continuous decline after sunset. This kind of misjudgement caused by multi-layer structure can also be seen from the lidar algorithm (Fig. 7(b)). Besides, the turbulence intermittency in SBL (Sun et al., 2015; Mahrt, 2014) could lead to the
underestimation of SBL. Under stable condition, the intermittency will cause discontinuous changes in meteorological elements and Fig. 8 shows one typical case. Three gradient methods determined the PBLH at 130 m, while the inversion in the profile of temperature extends significantly higher. According to the $Ri$ method, the PBLH at 2000 BJT on 14 April should be 468 m and that matches the continuous PBLH evolution in lidar profiles.

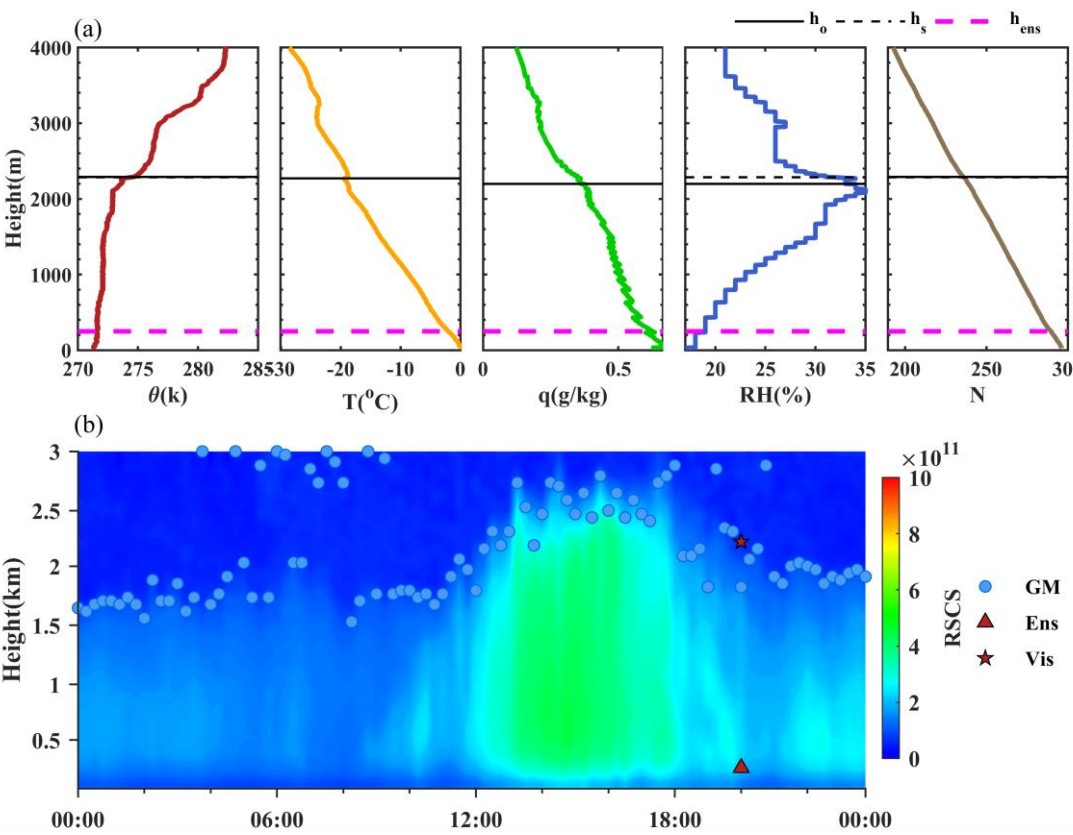

Figure 6: (a)The profiles of potential temperature ($\theta$), temperature ($T$), specific humidity ($q$), relative humidity ($RH$), and refractivity ($N$) at 2000 BJT on 9 February 2017. $h_o$ is the PBLH determined by original data, $h_s$ is the PBLH determined by data with three-point smoothing, and $h_{ens}$ is the PBLH determined by the ensemble method. (b) Evolution of the lidar RSCS signal at 1064 nm on 9 February 2017. The PBLHs retrieved from gradient method (GM) are marked by bule dots, the $h_{ens}$ in (a) is marked by a red triangle, and the PBLH of visual validation is marked by a res star.


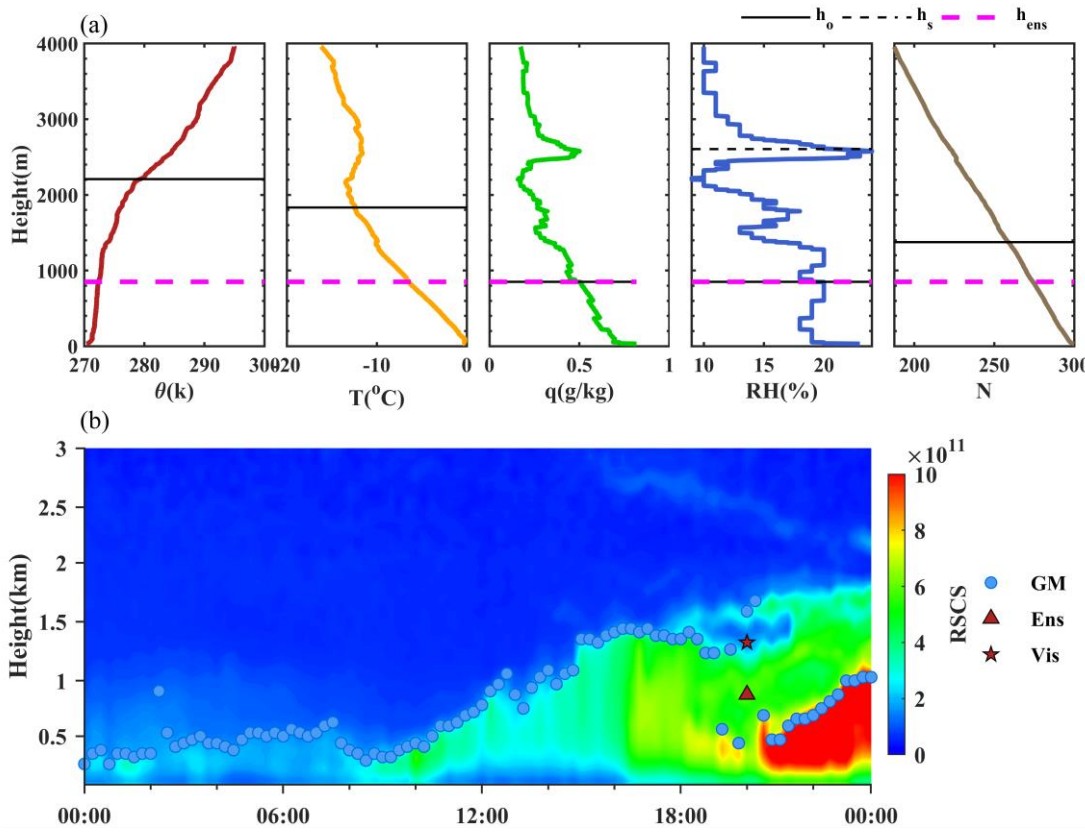

Figure 7: Same as Fig. 6 but at 2000 BJT on 14 January 2017.

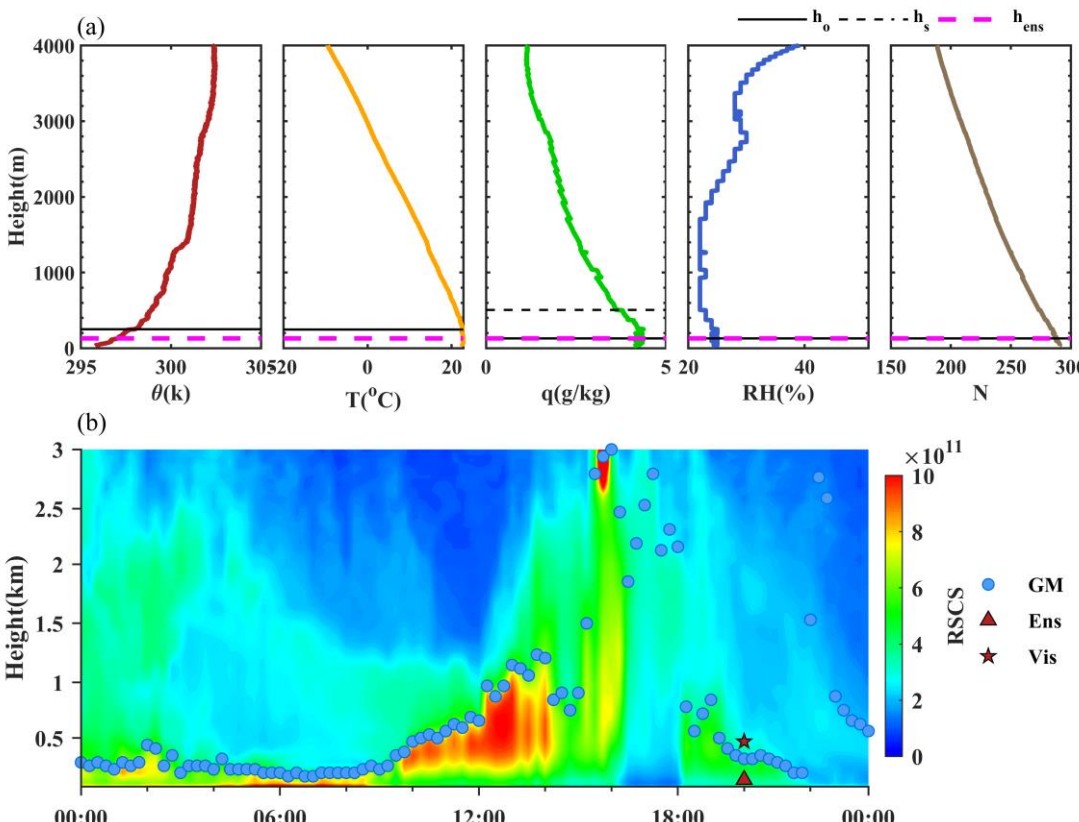

**Figure 8: Same as Fig. 6 but at 2000 BJT on 14 April 2017.**

**3.4 PBLH of Different PBL Regimes**

Finally, the ensemble method was applied to study the seasonal and diurnal characteristics of PBLH. According to the criteria of classifying the PBL regimes described in Sect. 2.4, the seasonal variation of the occurrence frequency of three major PBL regimes at two conventional launch times is illustrated with the average sunrise and sunset times in Fig. 9. The CBL generally occurs at 0800 BJT in spring (20) and summer (27) when the sunrise is earlier, and the surface can become warmer with a longer period of solar radiation. Conversely, the SBL occurs more frequently in autumn and winter at both times. The

occurrence of SBL in spring and summer indicates that the surface cooling can even maintain 2 h after sunrise. The TS occurred 204 and 260 times at 0800 and 2000 BJT, respectively. The TS occurrences are dominant in summer and spring at both times because the earlier sunrise accelerates the transition from SBL to CBL and the sunset at about 2000 BJT delayed the formation of SBL by the surface cooling.

The diurnal variation of PBL is preliminarily described by the observations in the morning, at midday, and in the evening (Fig.

9c). In respect of CBL, the PBLH is about 600 m higher at 1400 than 0800 BJT as the convection is most vigorous at noon. Even the CBL could maintain after sunset, the surface cooling can attenuate the convection and make the average CBL height shrink to 727 m at 2000 BJT. The phase peak of the TS height is also at 1400 BJT with an average of 1318 m. Comparing with





the evening TS, the morning TS height falls to about 700 m after a whole night of homogenous turbulence attenuation. The mean SBL height at 0800 and 20:00 BJT is 517 and 674 m, respectively. The existence of remaining misjudgments of TS
heights as SBL heights caused the overestimation and dispersion of the SBL height at 2000 BJT. Besides, two times of SBL occurred during midday (1400 BJT), which was also observed by Liu and Liang (2010).

The seasonal variation of PBLH observed at two routine times is presented in terms of the box plot in Fig. 9d. The average CBL height varies between 600 and 1000 m in different seasons and shows to be higher in spring. As mentioned above, longer periods of solar radiation favor the development of convection. In contrast to the CBL, the 50th percentile value of the SBL is
lower in summer with about 200 m. In spring and winter, the 50th percentile values of the SBL rise to about 300 m due to the stronger surface cooling. The SBL is more variable in autumn because of the remaining inaccuracy of PBLH estimation. The TS height lies about 900 m and there is no significant seasonal variation in the average, but it is more variable in autumn and winter.

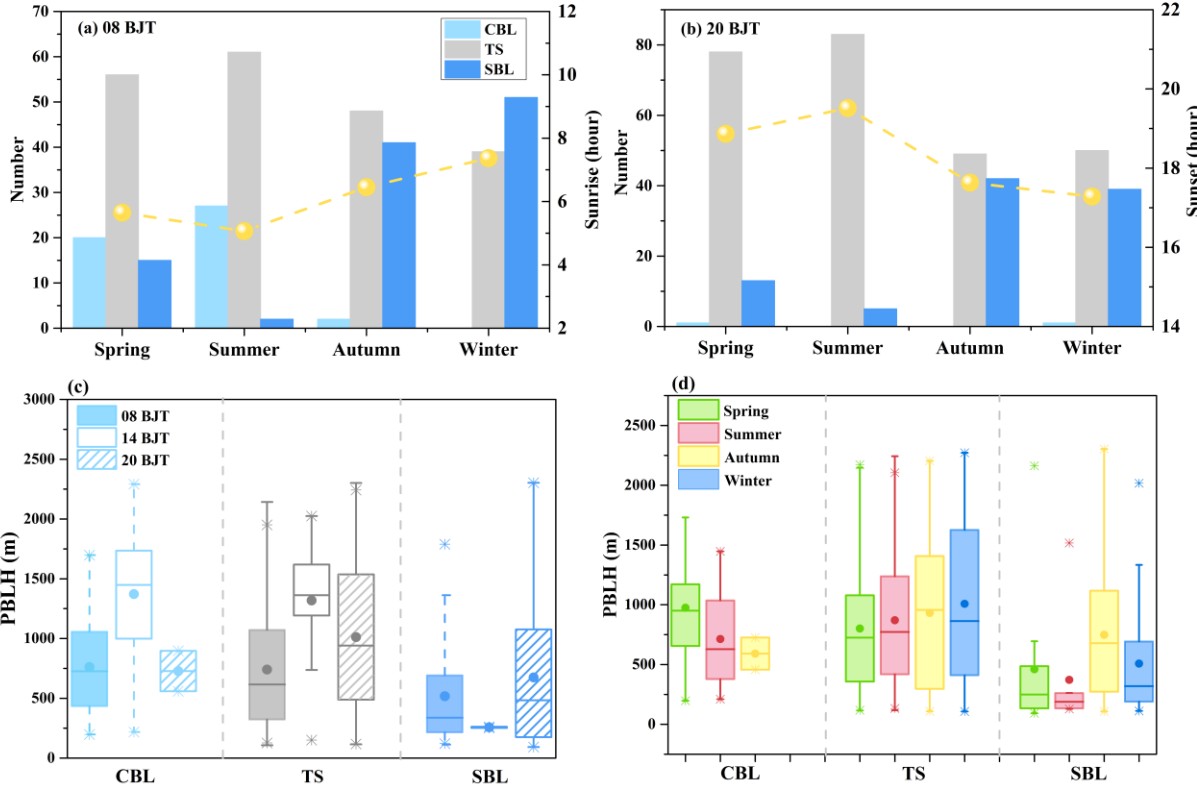

**Figure 9: The occurrence frequency of three PBL regimes in different seasons at two routine observation times of (a) 0800 BJT and (b) 2000 BJT. The yellow solid circles represent the average sunrise and sunset times for corresponding seasons. Box-and-whisker plots of three regimes of PBL at different (c) observation times and (d) seasons (only routine observations at 0800 and 2000 BJT are included). The dot in each box indicates the mean value of PBLHs and the cap represents the outlier.**



## 4 Conclusions and Discussion

In the present study, an ensemble method used to determine the planetary boundary-layer height (PBLH) from radiosonde was proposed to reduce the inconsistency between existing methods. Seven individual methods including four gradient-based methods, elevated inversion (EI) method, surface-based inversion (SI) method, and bulk Richardson number ($Ri$) method are combined along with the statistical modification. The ensemble method was applied to one-year high-resolution radiosonde data. Overall, the results show that the ensemble method has high potential to provide an accuracy estimation of PBLH for the

validation of other instruments, especially in the transition period.

    To first illustrate the effectiveness of the ensemble method, three typical cases during afternoon, morning, and evening transition periods are presented, respectively. The results confirmed that statistical modification of gradient-based methods can effectively eliminate the overestimation caused by the presence of clouds and the inconsistency between individual methods can be reduced by clustering. More validation was demonstrated by statistical analysis. Comparing with existing

methods, the annual average of PBLH from the ensemble method shows the best agreement with the refractivity ($N$) method and it is 676 and 917 m at 0800 and 2000 BJT, respectively. The comparable average PBLHs to existing methods also show the rationality of the ensemble method. Further visual proof of the effectiveness of each method was carried out for the one-year observations by comparing with lidar. The $Ri$ method shows the lowest effectiveness of 9.6% and the $N$ method shows the highest effectiveness of 56.1%. The ensemble method raised the effectiveness to 70.8%, which is 14.7% ~ 61.2% higher

than existing methods. Although the ensemble method shows a good improvement, there still some uncertainties in the calculation process and some cases where it is not applicable. The uncertainty of the ensemble method was evaluated by adjusting the threshold of statistical modification. The PBLHs of TS was most affected by the increase of 5% quantile with an increase of approximately 75 m. The SBL was least affected, but the difference of mean PBLH between 80% and 75% quantile is twice that between 75% and 70% quantile, which means a higher threshold could cause greater uncertainty. Besides, three

cases with the removal of truly high PBLH, the multi-layer structure, and the intermittent turbulence in the SBL are presented to make clear when the ensemble method is invalid. At last, the reasonable diurnal and seasonal variations derived from the ensemble method also indicate that the method is applicable for different regimes of PBL in the transition. The average CBL height shows to be the highest in spring and the SBL is lowest in summer. The average PBLH of TS is about 900 m and there is no obvious seasonal variation.

Generally, our method has been demonstrated to be effective. However, this method was only conducted at one typical station due to data limitations. Thus, detailed validations should be conducted at more stations in the future for further wide applications. On the other hand, this method still relies on the existing methods and that results in some shortcomings of these methods being retained. With the increase of more vertical profiles of turbulence observations, our improved understanding of the physical mechanism underlying the key physical and chemical processes in the boundary layer will help develop a better

method to estimate PBLH in a more realistic way.



**Author contributions**

The development of the ideas and concepts behind this work was contributed by all the authors. X.C. and T.Y. designed the research. X.C. and F.T.W. performed the formal analysis. H.B.W. was in charge of data curation. X.C. wrote the manuscript. Z.F.W., and T.Y. reviewed and edited the manuscript.

**Competing interests**

The authors declare that they have no conflict of interest.

**Acknowledgments**

We would like to thank the support from National Key Research and Development Program Young Scientists of China(Grant 2022YFC3704000)and National Natural Science Foundation of China under Grant 42275122. The author Ting Yang 370 gratefully acknowledges the Program of the Youth Innovation Promotion Association (CAS).

**Financial support**

This research is supported by the Strategic Priority Research Program of the Chinese Academy of Sciences (Grant No. XDA19040203); National High Technology Research and Development Program of China (No. 2019YFC214802); The Young Talent Project of the Center for Excellence in Regional Atmospheric Environment, CAS (CERAE201803).

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
