# Peer review of "An ensemble method for improving the estimation of planetary boundary layer height from radiosonde data"

_Atmospheric Measurement Techniques, 2023_

## Author Comment (AC1)

**Authors' responses to Reviewers' comments**

**Journal:** Atmospheric Measurement Techniques

**Manuscript Number:** AMT-2023-78

**Title:** An ensemble method for improving the estimation of planetary boundary layer height from radiosonde data

**Authors:** Xi Chen, Ting Yang, Zifa Wang, Futing Wang, Haibo Wang
* * *
**Anonymous Referee #1:**

**General comments:**

It has been well recognized that the estimate of planetary boundary-layer height (PBLH) from radiosonde varies dramatically by the methods used, especially at the morning and evening transitional period. To reduce the inconsistency between existing methods, this manuscript by Chen et al. proposed an ensemble method to confront this challenge based on one year worth of high-resolution radiosonde measurement at Beijing weather station. This algorithm has solid physical basis. The analysis methods are scientifically sound, and the results are reasonable from my point of view. The manuscript is well organized, and figures and tables are presented in a succinct way and easy to follow. Nevertheless, the effectiveness of this method seems elusive to me, and thus further clarification is needed. Therefore, I recommend this manuscript be accepted after a minor revision. The specific comments are as follows:

*Authors' response:*

We thank the Reviewer for her/his valuable comments and detailed corrections for our manuscript. We have considered the detailed comments and responded orderly as listed below.

**Major comments:**

1. "the effectiveness of the ensemble method" appears several times through the whole

manuscript. But I can not find the exact definition for EFFECTIVENESS. For the benefit of readership, necessary clarification for this noun is required. Therefore, the authors are suggested to give an unambiguous definition for EFFECTIVENESS with some explanation for its implication?

*Authors' response:*

We thank the reviewer for the valuable suggestion. We consider the result to be valid when the error between the result of the corresponding method and the PBLH with visual validation is less than 50 m. The effectiveness represents the proportion of the valid samples to all samples. The clear definition for effectiveness ($E$) has been added in section 3.2 to help readership comprehend. Besides, "effectiveness" is replaced by "$E$" in the rest of the manuscript.

Page 11, Line 250-252:" *If the error between the result of the corresponding method and the truth PBLH value is within 50 m, then the result will be considered valid.* **So, the effectiveness of each method is defined as following:**

$$E = \text{number of valid samples} / \text{number of all samples} \times 100\% \qquad (3) "$$

2. Figure 9: I do not understand what does the Y-axis on the right-hand side of panel (a) and (b). The authors can clarify this in the figure caption or in the main text.

*Authors' response:*

We are sorry for not specifying the axis information in Figure 9. The yellow solid circles in panel (a) and (b) represent the time of sunrise and sunset, respectively. We have changed the Y-axis label on the right-hand side to "Sunrise/Sunset (BJT)", and a more specific clarification

has been added in the figure caption.

[Figure]

*Figure 9:* **Histograms of occurrence number of three PBL regimes in different seasons at two routine observation times of (a) 0800 BJT and (b) 2000 BJT. The yellow solid circles in (a) and (b) represent the average time of sunrise and sunset in BJT and correspond to the right Y-axis.** *Box-and-whisker plots of three regimes of PBL at different (c) observation times and (d) seasons (only routine observations at 0800 and 2000 BJT are included). The dot in each box indicates the mean value of PBLHs and the cap represents the outlier.*

**Minor comments:**

L19: "during afternoon, morning, and evening transition periods, respectively." can be rephrased as "at 0800, 1400 and 2000 Beijing time"

We rephrased the sentence according to the suggestion.

L25: what does CBL stands for? For its first appearance, the acronym is supposed to be given

a full name.

CBL stands for convective boundary layer. We apologize for not defining the acronym when it was first used, and the revision has been made in the manuscript.

L27: It is better to use an adjective to describe the EFFECTIVENESS of the ensemble method developed in the present study.

According to the suggestion, the sentence has been rephrased as "These findings imply that the ensemble method is reliable and effective."

L36: "with the free troposphere" can be revised to "between the free troposphere and ground surface".

Changed according to suggestion. (Page 2, Line 36)

L47: the dash line in "wind-profiler" can be dropped.

Changed accordingly. (Page 2, Line 48)

L73: "As the routine radiosonde generally operates" -> "As the routine radiosonde measurements are generally taken"

Changed according to suggestion. (Page 3, Line 73)

L77: "further understanding of the transition period" -> "further understanding of the PBL structure and evolution during the transition period".

Changed according to suggestion. (Page 3, Line 77)

L238 and 245: "the integrated method" is used instead of "the ensemble method" that appear in the title of this manuscript. Are there any differences between them? If not, I suggest the authors use one term through the whole manuscript.

Thank you for pointing this out. There is no difference between "the integrated method" and

"the ensemble method". We have made revisions in the manuscript to make sure that only one term is used throughout the whole manuscript.

L334: "an accuracy estimation" -> "a reliable estimation"

Changed according to suggestion. (Page 17, Line 340)

L336: "first" can be removed.

Changed accordingly. (Page 17, Line 342)

L367-368: "and that results in some shortcomings of these methods being retained" can be rephrased.

This sentence has been rephrased as "On the other hand, some shortcomings of the existing methods may be retained in the ensemble method."

---

## Author Comment (AC2)

**Authors' responses to Reviewers' comments**

**Journal:** Atmospheric Measurement Techniques

**Manuscript Number:** AMT-2023-78

**Title:** An ensemble method for improving the estimation of planetary boundary layer height from radiosonde data

**Authors:** Xi Chen, Ting Yang, Zifa Wang, Futing Wang, Haibo Wang
* * *
**Anonymous Referee #2:**

**General comments:**

PBLH derived from radiosonde profile is a commonly used method and also taken as the standard for other method. However, there is usually big difference between the PBLH determined based on different variable profiles. An ensemble method based on high-resolution radiosonde data in Beijing is proposed to derive PBLH. The new method aims to decrease the uncertainty of PBLH estimation, but there are still some questions in the new method, which will increase the uncertainty of the results. The paper is recommended for publication after the following comments been properly addressed.

*Authors' response:*

We thank the reviewer for the constructive suggestions and the recognition of our work. We have considered these comments and responded orderly as listed below.

**Major comments:**

1. 75% quantile of the annual result from the specific gradient method is taken as the threshold in the 2nd step of the new method. As we all know, it indicates significant seasonal cycle of PBLH, which is usually highest in summer and lowest in winter. If taken the 75% PBLH of the

whole year as the threshold, the number of cases needed to be modified will be highest in summer, which will underestimate the derived PBLH in summer, and vice versa. I would like to suggest using seasonal or monthly threshold instead of the annual PBLH threshold.

*Authors' response:*

We really appreciate the valuable suggestion. Considering the number of profiles, we finally used seasonal threshold instead of annual threshold. The four seasons correspond to March to May, June to August, September to November, and December to February, respectively. The seasonal difference in threshold is shown in the following figure. For $\theta$ method, the seasonal variation is significant at 0800 BJT, with the highest threshold in winter (2381 m) and the lowest threshold in summer (1386 m). But at 2000 BJT, the seasonal variation is significant for RH method. The threshold shows a slight increase from spring to autumn, but it is significantly higher in winter (>400 m) for all methods. This indicates that the PBLH determination in winter may have greater bias. As the seasonal threshold is applied, the flow chart of the ensemble method (Figure 1) and the relevant descriptions in the manuscript have been modified accordingly.

[Figure]

The 75% quantile of gradient methods at (a) 0800 BJT and (b) 2000 BJT in different seasons.

[Figure]

Page 5, Line 131-133:" *2. Modify the results of gradient methods (θ, RH, q, and N). Get the 75% quantiles of initial results of gradient methods from the observations at 0800/2000 BJT for each season, respectively. Compare the result of each gradient method with the corresponding 75% quantile……"*

*Figure 1: The flow chart of the ensemble method.*

2. In the new method, if the initial result is greater than 75% quantile, then the highest level of the top 10 smallest gradients under 75% threshold is taken as the derived PBLH. That means if the derived PBLH higher than 75% threshold, it was a false result. Why cannot the derived PBLH higher than the 75% threshold? Based on your method, all of the final derived PBLH are lower than the initial 75% threshold, which will decrease the derived PBLH artificially. Could you give more explain about the 75% threshold from the perspective of atmospheric physics rather than mathematical statistics.

*Authors' response:*

We do agree with the reviewer. As we have pointed out in the original manuscript, the removal of truly high PBLH is one of the important reasons for the failure of the ensemble method. Therefore, we made a discussion of the uncertainty caused by the threshold in section 3.2 and suggested to get the climatology value from the previous research on the climatology of PBLH as threshold. Unfortunately, we only choose the 75% threshold from the perspective of mathematical statistics. We also pointed this out in the discussion:" …… *our improved understanding of the physical mechanism underlying the key physical and chemical processes in the boundary layer will help develop a better method to estimate PBLH in a more realistic way.*"

In order to avoid the truly high PBLH being eliminated, the ensemble method is modified in the manuscript. In step 2, a criterion was added before the statistical modification. We first calculate the difference between the initial result with all other methods and find the result with a difference less than 50 m. If at least one-third of differences are less than 50 m, then the initial result will be accepted, and the statistical modification will be stopped. Here, we give an example at 2000 BJT on 9 February 2017 to show the effect of the modification in step 2. In this case, all existing methods determined the PBLH at approximately 2200 m and this height exceeded the 75% quantile for $q$ and N methods. The statistical modification was initiated in the original algorithm, which resulted in the underestimation of the PBLH as 250 m. In the new method, more than a third of results are close to 2200 m, so the statistical modification is not enforced and the truly high PBLH is retained. The PBLH was finally determined at 2282 m, which is consistent with development of PBL observed by lidar.

[Figure]

(a)The profiles of potential temperature ($\theta$), temperature ($T$), specific humidity ($q$), relative humidity ($RH$), and refractivity ($N$) at 2000 BJT on 9 February 2017. $h_o$ is the PBLH determined by original data, $h_s$ is the PBLH determined by data with three-point smoothing, and $h_{ens}$ is the PBLH determined by the ensemble method. (b) Evolution of the lidar RSCS signal at 1064 nm on 9 February 2017. The PBLHs retrieved from gradient method (GM) are marked by bule dots, and the $h_{ens}$ in (a) is marked by a red triangle.

Page 5, Line 133-137:" ......If the initial result is greater than the corresponding 75% quantile, then get the difference between the result and other methods. Accept the result, if at least one-third of differences are less than 50 m. Otherwise, go through altitudes of the 10 smallest (or largest for θ) gradients and replace the initial result with the first altitude less than 75%

*quantile. If all altitudes do not meet the criteria, then the PBLH for the specific observation is*

*null."*

3. In the final step of the new method, the results are divided into several groups by 50 m, and the result of the group with the largest samples is used for the final PBLH estimation. Although seven variable profiles are used in this study, these variables can be divided into three groups, i.e., vertical temperature information group or humidity group or both of them. As show in Wang and Wang 2014, the vertical humidity observations usually suffer from the existence of clouds and the measurement error of humidity instruments, which is the main reason for the difference of PBLH between temperature profile group and humidity kind group. That means, it shows more consistency between the PBLH in temperature group or humidity group or temperature-humidity group. And the difference between the three groups is larger than those in the individual group. So, the number of temperature and humidity variables in the seven selected variables will influence the sample number of each group in the final step of the new method. If five of the seven variables are only related to humidity information, it will increase the probability of the five derived PBLH grouped together in the final step, which will lead to the missing of temperature information.

*Authors' response:*

We really agree with the reviewer's comment. So, we divided these methods into two groups and counted how often they were eventually adopted in the final step. One is temperature group, including potential temperature ($\theta$) method, elevated temperature inversion (EI) method, and surface-based inversion (SI) method. The other is temperature-humidity group, including

specific humidity ($q$) method, relative humidity (RH) method, refractivity (N) method, and bulk

Richardson number (Ri) method. The number of variables in the two groups is close, and neither

group is dominant. In the final step, 55.2% of the cases included the results from the temperature

group and 98.1% of the cases included the results from the temperature-humidity group. This

indicates that neither temperature of humidity information has been lost in the ensemble method.

Besides, we can see from Figures 4 and 5 that the difference between the two groups is

comparable to the difference in the individual group. Therefore, even though the effectiveness

of the temperature group is lower than temperature-humidity group, it is still important when

the difference in the temperature-humidity group is large.

4. It shows highly vertical resolution of the raw radiosonde data. Three-point smoothing was

conducted to avoid the influence of high resolution. I suggest to make a sensitivity analysis

about the effects of the number of smoothing points to the final result of derived PBLH.

*Authors' response:*

Following the reviewer's comment, we made a sensitivity analysis about the effects of the

number of smoothing points. As the PBLH error limit is 50 m and the average vertical resolution

is 5-8 m, only three-point smoothing, five-point smoothing, and seven-point smoothing were

applied to the ensemble method, respectively. The average PBLH is highest with five-point

smoothing (1159 m) and lowest with three-point smoothing (1109 m). According to the

comment from Reviewer #1, we also made a definition for effectiveness ($E$) in section 3.2

(Equation 3) to illustrate how effective the ensemble method is. The $E$ of three-point smoothing,

five-point smoothing, and seven-point smoothing is 62.6%, 59.6%, and 57.8%, respectively.

More smoothing points may cause some boundary layer structures to be lost, so the results are not more reliable. All the above results show that the PBLH from ensemble method is more sensitive to the number of smoothing points than the selection of threshold. So, the sensitivity analysis had been added in section 3.2 to complete the discussion of the uncertainty of the ensemble method.

Page 12, Line 267-270:*" ......, which means the uncertainty caused by the threshold is small. Furthermore, the number of smoothing points can be another source of uncertainty. We applied five-point smoothing and seven-point smoothing in the ensemble method, respectively. The increase in smoothing points increased the average PBLH by about 50 m and the difference between five-point and seven-point smoothing is small. More smoothing points may cause the loss of PBL structure, so $E_{7\text{-}point}$ (57.8%) is the lowest."*